# Applications and Properties of Hemp Stalk-Based Insulating Biomaterials for Buildings: Review

**DOI:** 10.3390/ma16083245

**Published:** 2023-04-20

**Authors:** Borja Martínez, Ernest Bernat-Maso, Lluis Gil

**Affiliations:** Department of Strength of Materials and Structures in Engineering, Polytechnic University of Catalonia, 08222 Terrassa, Spain; borja.martinez@upc.edu (B.M.); ernest.bernat@upc.edu (E.B.-M.)

**Keywords:** hemp stalk, bio-insulating material, sustainability, sound absorbing properties, thermal absorption properties, green composite material

## Abstract

There has been increasing interest in green and recyclable materials to promote the circular economy. Moreover, the climate change of the last decades has led to an increase in the range of temperatures and energy consumption, which entails more energy expenditure for heating and cooling buildings. In this review, the properties of hemp stalk as an insulating material are analyzed to obtain recyclable materials with green solutions to reduce energy consumption and reduce noise to increase the comfort of buildings. Hemp stalks are a low-value by-product of hemp crops; however, they are a lightweight material with a high insulating property. This study aims to summarize the research progress in materials based on hemp stalks and to study the properties and characteristics of the different vegetable binders that could be used to produce a bio-insulating material. The material itself and its microstructural and physical aspects that affect the insulating properties are discussed, as is their influence on durability, moisture resistance, and fungi growth. Research suggests using lignin-based or recyclable cardboard fiber to develop a bio-composite material from hemp stalk, but long-term stability requires further investigation.

## 1. Introduction

Energy consumption has been increasing in recent decades, as has its consequent increase in greenhouse gas emissions into the atmosphere. In the building sector, climate conditioning consumes 45% of the total energy consumption in the building [1]. This effect is increased every year as long as more and more energy is needed to keep ideal temperatures. Due to the new requirements, studies are carried out to improve the energy performance of buildings, either through passive or active techniques. The demand for green buildings, “a building that, in its design, construction or operation, reduces or eliminates negative impacts, and can create positive impacts, on our climate and natural environment,” [2] will increase substantially in the coming years. Through this type of technology, it is possible to reduce the carbon footprint of human activities and to achieve the objectives set in the Paris Agreements, to aim at “net-zero” emissions by 2050 [3].

A technique that can be explored in this field of research is the improvement of the performance of insulating materials based on green materials [4,5] with a positive carbon footprint [6]. Biomaterial or eco-friendly materials are materials that are produced and used in a way that minimizes harm to the environment. These materials are biodegradable, renewable, and sustainable, and have a reduced carbon footprint compared to traditional materials, reduce the amount of waste generated, and decrease the need for virgin materials. The use of eco-friendly materials is an important step towards achieving a more sustainable future. The European Industrial Hemp Association (EIHA) states that the hemp industry is an economically viable and socially responsible enterprise, as it contributes to restoring ecological balance and achieving decarbonization goals for a promising sustainable economy [7].

Hemp, *cannabis sativa* L., is a crop that can grow rapidly and yield up to four harvests in a single year. It is mainly produced for industrial or food purposes and is available in different varieties, each suited for specific applications. Hemp is a multi-functional crop with diverse applications across several fields. Despite its versatility, hemp cultivation declined globally in the 20th century due to the emergence of synthetic fibers that were cheaper to produce and possessed superior properties [8].

In recent years of the 21st century, hemp has aroused interest in the scientific community, considerably increasing the number of related publications in recent years (Figure 1). The figure shows the number of published articles only by ScienceDirect and MDPI to confirm the incremental tendency of scientific interest. However, the review takes into account articles published with other editors, considering the year of publication, methodology, and citations of each article.

Although it is primarily used in the textile industry, with the advent of the industrial era, other parts of the plant have been utilized in the market, leading to an increased range of products and applications [9]. These applications include fibers for textiles, seeds and oils for food, biomass fuel, construction materials, insulation for automotive textile parts, paper production from cellulose, and medicinal uses.

The remarkable versatility of hemp, increased farming productivity, rising demand resulting from emerging applications, and the growing need for eco-friendly materials have led to a surge in hemp farming in recent years. Despite the great farming tradition, new studies are still important to optimize the farming of hemp due to the numerous varieties of the crop. It is important to note that the usage and production of natural fibers are influenced by the customs and traditions of specific countries and regions. For example, China and Bangladesh, both located in Asia, have a rich history of the use of natural fibers, particularly jute, sisal, and coconut. Conversely, France and Belgium are renowned for being major producers of flax, whereas North America, despite its significant economic impact, does not possess the same level of prominence in relation to natural fibers. West Africa, Latin American countries, and India are the major oil palm cultivating countries [10,11,12,13]. In the case of hemp production, in Spain, according to data from the Ministry of Agriculture, in 2016, there were 61 ha devoted to hemp cultivation, and in 2020, that number increased to 510 ha [14]. This trend is also being reflected in the world, with Canada, the USA, China, and France being the countries with the highest number of hectares of hemp farms [15,16,17]. Industrial hemp production grew over 70% in Canada [18]. The US has increased its production from 9000 ha in 2016 to 93,000 in 2019 [19,20]. Furthermore, the estimation of the industrial hemp market presents gratifying results: the market size was estimated at $4.13b in 2021 and was expected to grow at a compound annual growth rate (CAGR) of 16.8% from 2022 to 2030 [21].

In addition to the innovations in the production of hemp, there are others related to less widespread applications, such as the use of fibers not only in the manufacture of textiles but also in construction, due to their mechanical properties when composite materials are manufactured from hemp fibers, obtaining a substitute material for synthetic fibers [22,23]. It also shows great potential in its use for biomass as a substitute for coal [18,24] or to produce activated carbon derived from hemp crops that can be utilized as an electrode material in a hybrid supercapacitor in order to produce a more environmentally friendly form of energy storage [25]. The reduction of CO_2_ emissions can also involve the use of hemp, which can be utilized for its negative carbon footprint. Moreover, hemp biomass can also be processed into biochar, which has the ability to absorb various types of organic and inorganic pollutants in an environmentally friendly manner [26].

However, currently, there is a part of the plant that does not have a specific application, and most of such material is treated as a low-value by-product of hemp crop or crop waste. Despite the fact that hemp stalk is used in building applications or to breed animals, those applications do not cover all the offers of the material [24]. The hemp stalk is localized in the inner part of the stem and represents more than 50% of the entire plant by weight [27]. Despite not having any specific application that satisfies all the offers, it has interesting properties as an insulating material. In addition, it is a biomaterial that contains cellulose and has woody fibers. Hemp fiber is one of the vegetable fibers that contain the highest percentage of cellulose (70–74%) [22,28]. Although the hemp stalk has a lower amount of cellulose than the fibers, the percentage is still considerable (Table 1).

The objective is to revalue a hemp by-product, hemp stalk (Figure 2) as the main material to develop a new biomaterial to use in green building will be reviewed, due to its insulating properties [33,34,35], positive carbon footprint, and its capability to produce a circular economy as a green material [36,37,38]. All these characteristics could be used to provide the stalk with new applications that add value to the material and, moreover, will increase the value of hemp farming [39]. So, in this way, more ecological materials will be produced.

The main focus of this review is to cover the properties of hemp stalks and different green materials that can be used as a binder to develop a green insulating material based on hemp stalks. Recently published papers indicate that lignin-based resin, bio-epoxy resin [5], and recyclable cardboard fiber [40] are potential binding materials for developing a new bio-composite material based on hemp stalk. While this new bio-composite material based on hemp stalk and eco-friendly binding materials shows promise as a replacement for traditional inorganic insulating materials in building construction, further research is needed to improve its long-term stability. It is important to ensure that the material maintains its structural integrity and insulating properties over time to ensure its effectiveness and durability in real-world applications. Continued investigation and development of this material will be crucial to its success as a sustainable building material.

## 2. Insulating Material

Thermal/sound-insulating materials are materials that prevent the transfer of heat or sound. In the case of a building, the insulating materials improve the performance of energy consumption and provide a more comfortable area. The most used materials are rock wool and EPS (Expanded Polystyrene).

One of the most attractive properties of hemp is its insulating properties [41]. Moreover, in comparison with conventional materials, natural fibers have similar hygrothermal properties, and the process of retrofitting historical building envelopes is made more harmonious by their involvement [42].

### 2.1. Insulating Properties of Hemp Stalk

Hemp possesses exceptional insulating qualities, which are retained in the stalk. One of the primary advantages of using hemp stalk is its affordability, coupled with its insulation performance. However, ensuring its long-term stability as well as its chemical compatibility with various binders is crucial [43]. Since hemp is a particulate material, its mechanical properties, as well as thermal and acoustic insulation capabilities, are dependent on the sample’s morphology. Therefore, the internal porosity of the material is the critical factor that determines its performance [44].

In the case of acoustic properties, the parameters of the hemp particles that influence its acoustic properties are the density of the particles, the bulk density, the thickness of the particles, and the shape factor [45]; this relationship is also shown in other biomaterials. On the other hand, the results also show that for low-frequency waves, acoustic insulation is not very effective due to the porosity of the material, although acoustic absorption can be increased in this range of frequencies by making a sandwich-type material.

Decreasing the stalk particle size improves the acoustic properties, the best case being those with an average length of 4 mm. Furthermore, the internal porosity can be predicted from the density of the particles [45,46]. At a microscopic level, hemp has a structure that brings on the absorption of acoustic waves due to the voids provided by the porosity of the material [33]. Hemp has a porosity of 78% with pores between 0.9–3 μm [47,48]. This study determines that the optimal properties for increasing the acoustic absorption of the stalk are for a particle to be 6 mm long with a density of 0.3 g/cm^3^.

In the case of thermal insulation, it increases energy efficiency by reducing heat transfer with the outside and saving energy in the air conditioning of buildings. The porosity also contributes to its high thermal properties, although in this case, the properties of the matrix composite material are more influential properties for the final property of the biomaterial composite.

According to the hemp stalk’s insulating properties, Table 2, hemp stalk is a green material that can be used in insulating applications, even though its properties are less competitive than commercial materials. The thermal conductivity of rockwool is 0.036–0.037 (W/m k) (28%, better performance than hemp stalk) [49,50] and EPS have 0.038 [50] (28%, better performance than hemp stalk). The acoustic absorption (α) of rockwool is 0.98–0.99 [51,52] (0.1 dB/dB, better performance than hemp stalk) and EPS 0.8–0.9 [53] (same performance of hemp stalk).

The elastic modulus of the hemp stalk (10–16 GPa) depends on its position in the stem of the plant according to the height at which it is located [56,57]. However, the area where the elastic modulus is greater is different among the species of hemp. The differences between the elastic modulus correspond to an increase or decrease in the size of the cell wall along the stem [56]. No studies have been found that prove whether these differences in cell size affect the insulating properties, detecting a research gap that could be the basis of interesting future research to increase hemp performance as raw material for insulating applications.

The properties shown by the stalk are suitable for use as both thermal and acoustic insulating material. However, it is a particle material, so it is necessary to use a vegetal binder to manufacture green composite material. In this insulating application, the mechanical resistance of the material is not a critical value; it is only necessary that it satisfy the minimum requirements and be a stable material.

### 2.2. Binder Materials for Hemp Stalk

This review focuses on green materials; however, there are not many studies with a 100% green material, so some results of non-green binder will be presented to study the behavior of the material and find out which green materials can obtain the best performance.

Different applications are being studied to use hemp stalk, taking advantage of its low price, such as using it as insulation in buildings and adding lime to form non-structural blocks [58,59]. Seeking a sustainable substitute for traditional walls, researchers carried out a study of the acoustic absorption properties of lime and hemp stalk walls, obtaining an average of between 40–50% acoustic absorption, obtaining better results with the less hydric binders and which can be manufactured on-site or placed as prefabricated units [60]. The main advantages of hempcrete are the insulating properties provided by the hemp stalk, and the lime binder provides protection against moisture, fungi, and fire [61]. In the same approach, adding hemp particles to the mortar as aggregates reduces its density, increases the insulating properties, and the material will increase the capability of CO_2_ storage. Nevertheless, the mechanical properties decrease (maximum stress is reduced up to 30% when adding 8% hemp) [62,63,64]. Although cementitious matrices offer the benefits of affordability and adaptability, their use can result in chemical damage to hemp stalk [65]. In hot-dry regions where naturally ventilated buildings are preferred, the thermal design of walls based on compressed earth blocks (CEB) could be a viable and more ecological option [66]. Moreover, the thermal behavior of CEB was improved by 10% by incorporating 0.5% of date palm waste [67]. While compacted earth is a compelling environmentally-friendly material, it is crucial to thoroughly examine its dependability and longevity in the absence of long fibers [68].

Hemp stalk particles are also used as raw materials for compounds that are made up of wood particles. In this way, the manufacturing process mixes it with a binder to fabricate a material similar to a chipboard. The manufacture of this type of material consists of mixing the stalk with the binder material and applying pressure and temperature in a mold, then the adhesive cures and the material obtains the shape of the mold. The most commonly used binders are currently based on formaldehyde because of its mechanical properties, dynamic properties, abrasion resistance, and affordability [69]. Nevertheless, due to the fact that it is a toxic material in large quantities, its use has been decreasing in order to reduce formaldehyde. An intermediate solution is the use of 2 formaldehyde-based adhesives by partially substituting them for lignocellulose-based materials (wheat straw and pine and poplar particles), obtaining better results with PDMI (Polymeric Diphenylmethane Diisocyanate), obtaining better results by increasing the percentage of binding material [70,71]. PDMI shows better binding properties than UF (Urea formaldehyde), curing at a temperature of 180 °C and having pressure applied for 3 min [72]; these are the usual values in the industry. The process is also the same with vegetable agglomerate taking into account the curing temperature for each vegetable binder [73,74]. In this type of manufacturing, the structure, and size of the particles is also an important factor in the final properties of the material. If the particle size is very large, air gaps will be produced in the material as all the chips cannot be compacted together because the manufacturing process does not use a vacuum to prevent the air gaps. However, this problem can be solved by including saws and wood dust that occupy these holes together with the resin, thus increasing the mechanical properties of the material [75].

Another alternative is to completely eliminate formaldehyde-based resins by using synthetic ones. Although studies are being carried out to obtain vegetable resins that can achieve the regulatory requirements of the different applications, such as lignin-based wood adhesives [76,77], obtaining materials with great thermal properties, or vegetable proteins such as camellia protein, which is also a residue in the biodiesel production [78]. However, the main problem is the resistance to fire; that problem can be solved by adding a fire resistance coating. There are also studies to develop a sustainable, high-performance, and flame-retardant wood coating based on a curing agent of ammonium hydrogen phytate (AHP) [79,80]. Starch is also a good biobased binder for wood particles; for example, cassava starch binder can be used to elaborate low-density particleboard with excellent performance [81]. The fungi resistance is low; however, citric acid can be added to improve the fungal degradation by 10% [82].

Different innovative fabrication methods and renewable materials for thermal insulating applications are studied. However, the technology still needs more research to have a competitive price and solve different technical difficulties in using materials like vacuum insulating panels, aerogels, or nanocellulose. The research also proposes the use of recycled paper fiber as an insulating material based on cellulose [83]. To bind the different hemp particles in a material that can be lightweight, a manufacture method is needed that does not require applying high pressure to provide the maximum porosity in the material. Paper pulp fiber is proposed, which also has great thermal properties, and cellulose paper waste has a thermal conductivity value of 0.046–0.054 W/m K [84]. Among the different paper fibers, cardboard is a great option due to the facility for recycling and the mechanical/binding properties. Cardboard fibers have a mechanical resistance four times greater than eucalyptus fibers, measured from the binding [85,86]. Moreover, the length of the fiber is longer (cardboard 2.7 mm in average length and eucalyptus 0.76 mm [87]). Several layers of corrugated cardboard were tested, obtaining thermal conductivity values of 0.053 W/m K and a reduction of up to 80 dB in acoustic waves using several layers of corrugated cardboard [5]. In that case, the main problems were durability and moisture/fungi/fire resistance.

Table 3 summarizes the comparison of the performance advantages and disadvantages of bio-based wood adhesives for the stalk. Regardless of the type of bio-based adhesive used, it is crucial to assess their potential for wood composite application and comprehend their interaction with wood. This evaluation can provide valuable scientific insights to guide the development of adhesives in terms of mechanical strength, water and moisture resistance, and thermal and acoustic properties.

### 2.3. Acoustic Insulating Properties of Materials Based on Hemp Stalk

When a surface is contacted by sound waves, the energy is distributed into three categories: incident, reflected, and absorbed energy. In architectural acoustic design, it is helpful to utilize an average absorption coefficient that is assumed to rely solely on the physical attributes of the material. The sound absorption coefficient of any material is determined by the angle at which the sound wave hits the material and the frequency of the sound [92].

In the case of a composite material based on hemp stalks. The mechanism by which the materials absorb sound energy mainly involves three physical processes. Sound-absorbing composite materials have small holes that allow sound waves to access their interior, causing gas flow and friction. It triggers the conversion of a portion of the sound energy into heat energy, which leads to sound absorption. In the case of hemp stalks, when sound waves hit them, the viscous effects between air cavities attenuate some of the sound energy, converting it into heat. Moreover, the sound-absorbing composite materials have the ability to absorb certain sound waves through their own vibrations. Due to the force between chain segments, unique hollow structure, and large specific surface of hemp stalks, sound energy is attenuated and converted into heat and mechanical energy during propagation, resulting in an effective sound-absorbing effect [33,93,94]. The internal porosity of the composite is a crucial factor to consider in the sound absorption mechanism of a low-density insulation panel. The presence of voids, inner and outer spaces, as well as the density and thickness of the composites, directly influence sound absorption [95,96]. Low-density materials with more open structures exhibit lower absorption at low frequencies, while denser structures show better performance at higher frequencies (above 2000 Hz) [92]. Several studies investigating sound absorption in porous materials have found a direct relationship between thickness and low-frequency sound absorption. Increasing the thickness of the material leads to an increase in sound absorption at low frequencies. However, at higher frequencies, thickness has an insignificant effect on sound absorption [97]. Nevertheless, a sandwich-type material can increase acoustic absorption in low frequencies. The low porosity and high density of the material’s surface cause sound waves to be reflected [44].

Table 4 shows the results obtained in different studies:

The acoustic absorption values in Table 4 are lower than those presented by the hemp particles, which means that the composition of the binder and the new internal microstructure is a more significant parameter than the internal porosity or the particle size of the hemp stalk.

Based on the results presented in Table 4, it can be observed that a highly porous material provides good acoustic insulation as it contributes to the dissipation of sound waves, which is a more significant characteristic than sound wave reflection. Hemp stalk has been studied as an insulating material, but there are few studies on hemp due to the materials primarily used with inorganic binders that require the application of pressure and temperature to improve mechanical resistance, resulting in a denser material with low porosity that impairs its properties [103]. However, although a porous material performs better, a non-porous surface coating can be added to the sample to improve sound wave reflection. The use of certain binders can result in an elastic behavior of the composite material, allowing acoustical vibrations to be transmitted to the solid matrix. These elastic effects can have a significant impact on acoustic performance, causing both global effects on transmission and local effects on absorption [100].

### 2.4. Thermal Insulating Properties of Materials Based on Hemp Stalk

The main objective of thermal insulation is to enhance energy efficiency by limiting the transfer of heat through the building envelope. Insulation materials are designed to conduct heat poorly in order to minimize heat loss [104]. Heat conduction occurs due to the interaction between particles in a substance (solid, liquid, or gas) that results from particle movement. Hence, heat moves from more energetic particles to less energetic ones. Convection, on the other hand, refers to heat transfer between a solid surface and a fluid in motion, whereby heat is transferred through a combination of conduction from the solid to the fluid and bulk movement of fluid particles. Thermal insulation offers high thermal resistance, thereby retarding heat flow, primarily due to gases entrapped within the porous material structure [105]. Thermal insulation materials typically have low densities, which translates to high porosity. The insulation effect is largely attributed to the low thermal conductivity of still gases trapped within the voids of porous material [105]. The principal factors that affect thermal conductivity include raw materials, temperature, porosity, moisture content, and density. Other factors are airflow velocity and thickness [104]. For cellulose-based insulating materials, factors such as temperature, moisture content, and mass density are critical in determining the thermal conductivity value [42,83]. So, in this case, the environmental conditions (temperature, humidity) affect its insulation capacity.

In addition to acoustic insulation, the hemp stalk also has high thermal insulation properties, obtaining a coefficient of thermal conductivity of 0.05 W/m K (Table 2). In Europe, according to the DIN 4108, materials with a λ value lower than 0.1 W/m K may be classed as thermal insulating materials. Additionally, materials with thermal conductivity values lower than 0.03 W/m K are deemed highly effective as thermal insulators [104]. Moreover, its resistance to fire must be classified as at least one type E material according to the regulations. This means that the material should be able to withstand the attack of a small flame for a brief period without significant propagation of the flame. [106].

There are studies on that topic to develop new biomaterials using starch as a binder [107]. Straw and a geopolymer can be used to form an insulating material, obtaining thermal conductivity values of 0.101 W/m K [108,109], the biggest drawback being the resistance to water and fire when using green materials. Other studies show materials made with corn particles and epoxy resins obtain sufficient thermal conductivity values to be considered insulating materials [44].

Table 5 shows the results obtained in different studies:

In addition to using vegetable resins, it is also proposed to replace the particles of pine and other common trees with crop waste, such as hemp stalk. The influence of the starch-stalk ratio on the properties of the material is studied, and it shows that by increasing the hemp particle ratio, the mechanical properties are reduced due to the increase in porosity, which decreases the load transfer capacity. Nevertheless, it improves thermal performance [101]. The thermal insulating performance of materials is improved by higher porosity and density. Unlike acoustic properties, the thermal properties of the matrix material play a crucial role in determining the final properties of the composite material.

At a mean temperature of 24 °C, the apparent thermal conductivity of hemp stalk was tested for various densities and found to increase with rising temperature. While wood-based fiberboards are utilized as thermal insulation materials due to their low density and high thermal resistance, their porous internal structures make them sensitive to environmental changes [116]. Consequently, their thermal conductivity increases by roughly 50% as the temperature increases from 10 to 60 °C [117].

An increase in relative moisture in materials can lead to a decrease in their thermal conductivity and make them more prone to mold formation. For composites made of jute, flax, hemp shives, and fibers, a relative air humidity of 70% leads to a relative material humidity of 5–10%. Additionally, it has been demonstrated that an increase in material moisture from 0–10% results in an increase in thermal conductivity [118].

### 2.5. Carbon Storage Properties

Although the different studies show high insulating properties, they are not the only important characteristic of these materials. Developing a biomaterial based on hemp stalk can increase the value of the hemp crop, produce more environmentally friendly materials, and also result in materials that act as CO_2_ accumulators [2].

The hemp absorbs CO_2_; meanwhile, it is growing through the photosynthesis process. Moreover, due to very rapid growth, the biomass accumulated by hemp crops has a significant effect in absorbing atmospheric CO_2_; however some of the carbon stored in this biomass is turned back into the atmosphere as CO_2_ due to the biodegradation of leaves and roots. Assuming a concentration of 0.5 kg of carbon per kg of dry matter, it can be calculated that 1.84 kg of CO_2_ is sequestered per kg of dry hemp through photosynthesis during the plant’s growth. This means that a ton of dry hemp can store 325 kg of CO_2_, taking into account the amount of emissions during the farming as the diesel consumption, transportation of the seeds, etc [119].

Hempcrete is a sustainable building material made from hemp stalks, lime, and water, and the lime in the material also takes part in the CO_2_ sequestration due to the carbonation process of the lime. During the carbonation process, the lime reacts with the CO_2_ present in the air, resulting in the conversion of calcium hydroxide (Ca(OH)_2_) into calcium carbonate (CaCO_3_) [120]. The carbonation increases the mechanical resistance of the hempcrete, and, additionally, the absorption of CO_2_ during this process can have significant implications for the environmental impact of this product [121]. Hempcrete can store 300 kg of CO_2_ per m^3^ [122,123,124]. With these two properties, we have a new material that reduces the energy consumption of the building due to the thermal insulation capacities and also stores CO_2_, obtaining a material that would passively help to obtain buildings with zero emissions [125].

Based on these examples, it can be concluded that the binder used in the composite material has a significant impact on its carbon footprint. The use of recyclable materials with low impact on the carbon footprint is proposed for the binder material.

### 2.6. Durability

In order to develop new biocomposite insulating materials, it is necessary to ensure the durability conditions to ensure the commercial product can be reached. However, there are only a few papers that study the durability of long-term tests in those materials. The longevity of bio-composites can be impacted by various forms of biological degradation, such as mold growth, as well as environmental factors like fluctuations in temperature and humidity [105].

Investigating the water absorption characteristics of biocomposites is crucial, given the weak water resistance of biomaterial. For biocomposites used outdoors in construction applications, their water absorbency is a critical factor affecting their mechanical properties and dimensional stability [126].

Hemp fibers also present low stability against contact with water, which sees their Young’s modulus reduced by 50% after immersion in water [127]. Moreover, using a 100% vegetable material increases the risk of mold growing in high humidity conditions [115]. Mold growth can be relieved by increasing the pH of the material, as is the case with hempcrete, which has anti-fungal properties [61].

In the case of VFRCM (vegetal fabric reinforced cementitious matrix), the issue arises from the alkaline hydrolysis resulting from the production of Ca(OH)_2_ during cement hydration. The calcium adsorption is pH dependent, and the pectin contained in fibers can react with calcium ions in an alkaline environment [128]. Moreover, a reduction in compressive strength has been observed in hempcrete when using Ca(OH)_2_ treated hemp stalks compared to untreated ones [129,130]. To incorporate a material that is rich in Al_2_O_3_ and SiO_2_ can mitigate the degradation of the hemp fibers. This material should have the ability to consume Ca(OH)_2_ during mortar hydration [131]. Alternatively, the fiber can undergo physical or chemical treatment. To extend these findings to vegetable binders, it is crucial to ensure compatibility between the hemp stalk and the vegetal binder to achieve optimal results.

To prevent the degradation of the biomaterials, a solution is to introduce a coating in order to protect the material against environmental effects. To increase the water resistance of these materials, surface coatings such as NaOH, silane, and epoxy have been studied, which reduce the moisture absorption of the fibers [132,133,134,135].

In one research work, a vegetal coating is proposed by coating a bamboo particle with pine resin, obtaining a hydrophobic material after treatment; in addition, after the coating, bamboo particles become a stable material against the environment [136]. Nevertheless, the pine resin worsens the fire resistance properties. Arabic gum also presents good results as a coating in cases of humidity and fire, although it cannot be used in direct contact with water since Arabic gum is soluble in water, so when the layer is in direct contact, the protective coating is removed until the material is uncoated [137]. Although these coatings increase the durability of the material, no studies have been found that investigate how the insulating properties behave when these coatings are added.

The coating would not only have the function of protecting the material, but also of stabilizing the interior moisture. Since moisture has a great impact on the insulating properties, as seen in the previous sections.

## 3. Conclusions and Further Research Interests

According to the results presented in the different studies of the last decades, hemp is positioned as a green material with great insulating properties that could replace some inorganic commercial materials to improve the insulating performance of buildings and build green buildings. These data can be corroborated by the increase in interest in the scientific community, with the increase in research each year, and also with the increase in the hemp industry in the business sector.

Varieties and positions on the stem can cause differences in the elastic modulus of the hemp stalk of up to 60%. However, no studies have been found to confirm whether these differences in cell size impact the insulating properties of hemp.This research gap presents an interesting opportunity for future studies to explore ways to enhance the performance of hemp as a raw material for insulating applications. With the farming of the different varieties of hemp, it is necessary to study which variety produces the hemp stalk with the best insulating properties and in which part of the stem it is localized.It is suggested to use a binder that aids in creating a low-density and porous material as the microstructure of the composite material plays a significant role in determining its insulating properties.It is still necessary to delve into some issues in order to develop new green insulating materials, such as the study of more green binders that do not need to apply pressure or temperature during the manufacturing process. In this way, the manufacturing cost is reduced, and lightweight materials with high porosity would be produced that would stimulate the insulating properties of the composite material.In this way, paper pulp-based binder shows great opportunities for the development of future research.Despite all the studies, further research is still needed on the use of a 100% plant-based composite material, including the binder. However, among those presented, starch stands out.By using starch as a binder, an acoustic absorption (α) of 0.7 and a thermal conductivity of 0.03 W/m K have been achieved, which are respectively 0.2 db/db lower and 15% higher than conventional materials.It has been observed that only a few studies have investigated the acoustic properties, fire resistance, fungi growth, and long-term durability of 100% green composite materials.A ton of dry hemp can store 325 kg of CO_2_.Hemp is exhibited as a material capable of storing CO_2_ and producing a renewable material with a circular economy. The binder also affects the carbon footprint of the composite material. To prioritize the carbon footprint, a recycled biomaterial such as cardboard fiber can be chosen as a binder.It is necessary to protect the biomaterial against ambient conditions.The main problem for a biocomposite material is degradation over time. So in order to produce a commercial product it will be necessary to ensure the stability of the material in ambient conditions. Nevertheless, no research has been found that investigates the influence of using an eco-friendly coating on the insulation properties of a composite material.To protect the biocomposite material against ambient degradation, some vegetable coating, such as colophony, shows a great performance. Arabic gum is proposed as an effective solution for fire protection, but it only provides protection against moisture and not direct contact with water.

In this review, a large amount of research related to the development of new hemp stalk-based composite materials has been compiled. Much research has been done, and it is arousing increasing interest in the scientific community for its performance and sustainability reasons. Therefore, it is recommended that these issues continue to be investigated in order to develop a new competitive material capable of replacing current inorganic materials.

## Figures and Tables

**Figure 1 materials-16-03245-f001:**
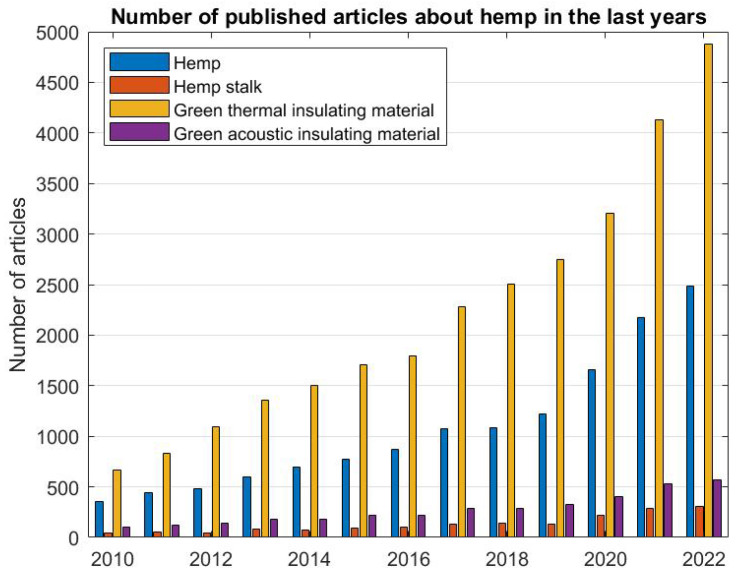
Number of articles published in ScienceDirect and MDPI about hemp and green insulation materials.

**Figure 2 materials-16-03245-f002:**
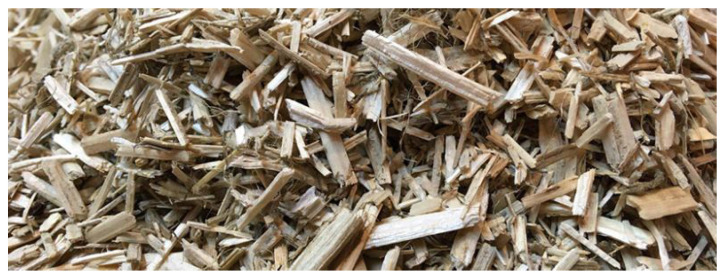
Hemp stalk.

**Table 1 materials-16-03245-t001:** Percentage of cellulose contained in hemp stalk.

Cellulose (%)	Hemicellulose (%)	Lignin (%)	Reference
44	18	28	[29]
50–60	15–20	20–30	[30]
34–44	31–37	19–28	[31]
49	25	25	[32]

**Table 2 materials-16-03245-t002:** Properties of hemp stalk.

Thermal Conductivity (W/m k)	Acoustic Absorption (α)	Density $\rho_{\mathrm{shives}}$ (kg/m^3^)	Porosity $\phi_{\mathrm{inter}}\left(\%\right)$	Reference
-	0.7–0.95	80–160	65–85	[45,46]
0.049–0.082	0.88–0.95	110–125	40	[47]
0.064–0.115	0.88–0.99	97–120	-	[54]
0.051	-	72	70–80	[55]

**Table 3 materials-16-03245-t003:** Advantages and disadvantages of bio-binder for stalk [5,76,77,78,79,81,82,83,84,85,86,87,88,89,90,91].

Type of Bio-Binder	Advantages	Disadvantages
Lignin based	Recycle the secondary products produced in paper pulping industries	Need to add a catalyst material
	Improve the modulus of elasticity	Increase the viscosity of adhesive
	Improve the thermal properties	Low fire resistance
	Improve the water resistance	Reduce the curing rate
	Good bonding strength	Low porous structure
		Low level of substitution
Starch based	High level of substitution	Low stability upon time
	Good bonding strength	Need a surface treatment to increase the water resistance
	Good film formation property	Low fire resistance
		Slow drying process
		Poor water resistance
		Low fungal resistance
Plant protein based	Improve thermal stability	Poor water resistance
	Good adhesion strength	Need a surface treatment to increase the water resistance
	High level of substitution	Low porous structure
		Low fire resistance
Paper pulp based	Improve thermal/acoustic properties	Need a surface treatment to increase the water resistance
	Good bonding strength	Slow drying process
	Recyclable	Poor fire resistance
	High porous structure	Low stability upon time
	High level of substitution	Low fungal resistance

**Table 4 materials-16-03245-t004:** Results of acoustic absorption of vegetal particles with different binders.

Fiber	Matrix	Eco-Friendly Material	Acoustic Absorption (α)	Pore Structure	Reference
Hemp stalk	Polycaprolactone	NO	0.6–0.9	Hollow microstructure	[33]
Hemp stalk	Lime	NO	0.6–0.9	Porous material (70–75%)	[46,60,98]
Hemp stalk	Portland cement & MgO-cement	NO	0.1–0.25	Low porosity	[99]
Hemp stalk	C2-H	NO	0.6–0.8	Porous material	[100]
Hemp stalk	Wheat starch	YES	0.7	Porous material (88–90%)	[89,101]
Sunflower stalk	Chitosan	YES	0.2	Low porosity	[102]
Sheep wool	Polypropylene	NO	0.3–0.6	Low porosity	[96]

**Table 5 materials-16-03245-t005:** Results of thermal absorption of vegetal particles with different binders.

Fiber	Matrix	Eco-Friendly Material	Thermal Conductivity (W/m K)	Reference
Hemp stalk	Lime	NO	0.08–0.13	[60,110,111]
Hemp stalk	Portland cement & MgO-cement	NO	0.08–0.115	[98,99,112]
Hemp stalk	Wheat starch	YES	0.06–0.07	[89,101]
Hemp stalk	Cassava starch	YES	0.026	[113]
Hemp stalk	Reactive vegetable protein	YES	0.078	[114]
Sunflower stalk	Chitosan	YES	0.056–0.058	[102]
Corn stalk	Rice huck ashes	YES	0.06–0.08	[107]
Bamboo powder	Bio-glues	YES	0.10–0.20	[115]
Corn stalk	Epoxy	NO	0.10	[44]
Sheep wool	Polypropylene	NO	0.06–0.10	[96]
Flax stalk	Lignin & and biobased epoxy	NO	0.074	[77]

## Data Availability

All the data is available within the manuscript.

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
