# Peer review of "Applications and Properties of Hemp Stalk-Based Insulating Biomaterials for Buildings: Review"

_materials, 2023, doi:10.3390/ma16083245_

Round 1
Reviewer 1 Report
The manuscript entitled “APPLICATIONS AND PROPERTIES OF HEMP STALK-BASED INSULATING BIOMATERIALS FOR BUILDINGS: REVIEW” provides some good results. Therefore, the current manuscript could be accepted for publication, but after going through a minor revision.
1. English language must be significantly improved.
2. The authors must explicitly state the novelty of their work in the abstract or at the end of the introduction section.
3. Why did the authors focus on MDPI and Science direct publications in figure 1? Other articles published by other publishers should also be included.
4. A clear cost analysis of should be provided by the authors.
5. Advantages and disadvantages of all of the used materials should be tabulated.
6. Some examples of recently published articles that could be useful for enriching the introduction section are:
https://doi.org/10.1016/j.mtcomm.2022.105207
https://doi.org/10.1016/j.jenvman.2022.115238
https://doi.org/10.1007/s10570-022-04773-6
7. More numbers and percentages that represent the major facts about the topic of this work should be provided in the conclusion section.
Author Response
Please, find enclosed our comments and modifications. Faithfully yours.

Reviewer 2 Report
Dear Editor,
Thank you for inviting me to review the article "Insulating Materials Based on Hemp Fibers for Buildings: A Review" by Martínez et al. After careful consideration, I believe that this manuscript has potential for publication after some corrections and improvements.
In my opinion, the authors should avoid using the term "hemp stalk as an insulating material" in the abstract, as it suggests that the review focuses solely on one product derived from hemp fibers, although the study dealt with different ways of applying these fibers in insulation materials for buildings. In fact, the valuable hemp fiber, used for various applications including textiles, ropes, paper, building materials, and more, is extracted from the stem/stalk of the plant. Therefore, I disagree with calling this material "waste crop material." Some sentences presented in the introduction of the manuscript supports my opinion.
The overall quality of the text needs improvement, as some phrases do not make sense. For example, "Among those that stand out the use of fibbers in the textile industry, seeds and oils in the food industry, as material for biomass fuel, construction materials, insulating materials for textile parts in automobiles, paper manufacturing from cellulose, medicinal applications, etc." It is unclear whether the authors meant to say that hemp was replaced by synthetic fibers or the opposite.
Regarding the introduction, it is known that the production and consumption of natural fibers depend on the local culture of certain countries and continents. For instance, Asian countries such as China and Bangladesh have an established culture of using natural fibers, especially jute, sisal, and coconut. On the other hand, France and Belgium are known as important producers of flax, while North America, although having great economic importance, does not have this prominence in the context of natural fibers. A description of this scenario concerning hemp fibers would be important to include in the introduction.
In section 2, I suggest including "earth blocks", "adobe bricks" or similar materials. These types of materials have gradually gained attention in the sustainable products market and are a great opportunity for those interested in applying hemp fibers.
In Table 5, the meaning of the term "biomaterial" is not clear. In fact, the literature defines this term as a material applied to the health sector, such as a prosthesis or dental material, but some people also understand a biomaterial as an environmentally friendly material. It would be interesting to comment on this terminological inconsistency and define what is meant in this manuscript.
I suggest implementing a standardization for reporting units throughout the manuscript. Sometimes units are abbreviated, while other times they are written out in full. Sometimes there is a space between the unit and the numeral, and other times there is not. There are also incorrect units, such as Kg, which should be kg, using a lowercase k.
It is not clear to me how the material called "hempcrete" can contain a CO2 content similar to that of a hemp fiber itself. This material involves the addition of cement, which has a high carbon footprint. It would be important to discuss this topic further, as little information was presented on this subject.
In the case of the durability of hemp fibers incorporated into cementitious materials, the problem lies in the alkaline hydrolysis caused by the generation of calcium hydroxide during cement hydration. Typically, this damage can be reduced by adding a material rich in silicates or aluminates, which must be capable of consuming calcium hydroxide during cement hydration, or by physical or chemical treatment of the fiber. This topic should be discussed in more detail.
The conclusions seem to me very ambiguous. The authors mention the different fiber variability, but do not indicate any preferred fiber type or comparison between fiber varieties. An issue related to the durability of products based on this fiber is also mentioned, but there is no indication of a preferential or good practice to mitigate this negative characteristic. Which treatments are most promising?
The last sentence of the conclusion suggests that a single competitive material capable of replacing inorganic materials is under development, which is obviously not true. I suggest improving this sentence.
Author Response

(The authors gave the same response as above.)

Reviewer 3 Report
In order to improve the paper quality, my suggestions and comments are shown below-
1. The authors need to incorporate the highlights of the study after the introduction section which can help to understand the review analysis.
2. Please cite the latest references related to your study in the introduction section.’
3. If possible add some other thermo-physical properties of the hem stalk materials in Table 2.
4. The conclusion need to be re-writing again to present actual finding related to study.
The overall manuscript is well written with collective manners, but it requires some minor correction, as discussed above. So I recommended minor revision before it accepts for publication.
Author Response

(The authors gave the same response as above.)
